# Nutraceuticals in the Prevention of Viral Infections, including COVID-19, among the Pediatric Population: A Review of the Literature

**DOI:** 10.3390/ijms22052465

**Published:** 2021-02-28

**Authors:** Giuseppe Fabio Parisi, Giuseppe Carota, Carlo Castruccio Castracani, Mariarita Spampinato, Sara Manti, Maria Papale, Michelino Di Rosa, Ignazio Barbagallo, Salvatore Leonardi

**Affiliations:** 1Pediatric Pulmonology Unit, Department of Clinical and Experimental Medicine, University of Catania, Viale A. Doria 6, 95125 Catania, Italy; gf.parisi@policlinico.unict.it (G.F.P.); saramanti@hotmail.it (S.M.); mariellapap@yahoo.it (M.P.); leonardi@unict.it (S.L.); 2Department of Biomedical and Biotechnological Sciences, University of Catania, Via S. Sofia, 87 95125 Catania, Italy; giuseppe-carota@outlook.it (G.C.); mariaritaspampinato93@gmail.com (M.S.); chitotriosidase@gmail.com (M.D.R.); 3The Children’s Hospital of Philadelphia (CHOP), Department of Pediatrics, Division of Hematology Leonard and Madlyn Abramson Pediatric Research Center, Philadelphia, PA 19104, USA; CASTRUCCIC@chop.edu; 4Department of Drug and Health Sciences, University of Catania, Viale A. Doria 6, 95125 Catania, Italy

**Keywords:** nutraceutical, children, viral infections, COVID-19, SARS-CoV2, probiotics, zinc, vitamin, resveratrol, quercetin, hesperidin, lactoferrin

## Abstract

In recent years, there has been a growth in scientific interest in nutraceuticals, which are those nutrients in foods that have beneficial effects on health. Nutraceuticals can be extracted, used for food supplements, or added to foods. There has long been interest in the antiviral properties of nutraceuticals, which are especially topical in the context of the ongoing COVID-19 pandemic. Therefore, the purpose of this review is to evaluate the main nutraceuticals to which antiviral roles have been attributed (either by direct action on viruses or by modulating the immune system), with a focus on the pediatric population. Furthermore, the possible applications of these substances against SARS-CoV-2 will be considered.

## 1. Introduction

On 30 January 2020, the World Health Organization (WHO) declared the COVID-19 pandemic a Public Health Emergency of International Concern (PHEIC) [1]. Severe Acute Respiratory Syndrome Coronavirus 2 (SARS-CoV-2) infects upper respiratory tract epithelial cells manifesting mild symptoms. Severe pneumonia can develop when the virus enters alveoli of the lungs and infects them, leading to the development of respiratory failure and acute respiratory distress syndrome (ARDS) [2,3]. In severe cases of COVID-19, the virus can enter the bloodstream and infect endothelial and other target cells in the kidneys, esophagus, bladder, ileum, heart tissues, and central nervous system. Patients in critical condition often develop “cytokine storm”, an elevated systemic inflammatory parameter involving levels of C-reactive protein, cytokines Interleukin (IL)-6, tumor necrosis factor (TNF)α, IL-8, etc. [4,5]. COVID-19 infection develops an initial pulmonary phase that consists of virus replication and inflammation, with the involvement of reactive oxygen species (ROS) as a direct inflammasome activator, and subsequent blood dissemination, possibly associated with the adaptive immune response to oxidative stress as a mechanism of systemic injury [6,7]. Thus, oxidative stress plays an important role in the pathogenesis of COVID-19, leading to the cytokine storm and blood clotting mechanism, and it exacerbates hypoxia linking an important participation of oxidative stress in the pathogenesis of viral infection in all direct tissue injury, which terminates in hypoxia and organ failure. Moreover, it reveals that the SARS-CoV-2 virus may interfere with the equilibrium between the nuclear factor kappa-light-chain-enhancer of activated B cells (NF-κb) transcription molecule involved in the expression of cytokines and nuclear factor erythroid-2-related factor 2 (Nrf2) activation, responsible for the expression of antioxidant enzymes [8,9,10,11]. Fortunately, the clinical manifestations in children appear to be much less severe than in adults [12,13,14,15,16]. However, severe cases have also been reported among younger age groups [17,18]. Beyond the classic therapeutic approach (not the objective of this work) and the common and well-known rules aimed at reducing the spread of the virus (masks, social distancing), there has been debate about the possibilities of preventing infection through the use of substances, such as nutraceuticals, capable of inhibiting the replication of SARS-CoV-2. These observations derive mainly from evidence showing that some of these substances are effective, more generally, in protecting against respiratory viruses. Nutraceuticals are those nutrients contained in foods that have beneficial effects on health. Nutraceuticals can be extracted, used for food supplements, or added to foods. It is rare to find nutraceuticals in natural foods in sufficient quantities to obtain these benefits [19]. Moreover, in the pediatric field, the term nutraceutical summarizes the concept of how an appropriate diet enriched with biologically active elements allows us to obtain important health benefits beyond the normal nutritional effects [20,21]. Therefore, it is possible to define as functional foods those foods that, due to their natural prerogative or supplementing, can provide vitamins, mineral salts, fibers, and fatty acids in such quantities as to positively influence specific functions or avoid the onset of diseases. Given this, two main classes of nutraceuticals can be identified: those that contribute to optimizing the growth processes and, therefore, the formation of tissues and systems; and those that have a strong preventive value, for example, against infectious episodes or pathological events [22,23]. Historically, many nutraceuticals have been attributed beneficial properties towards human health [24,25]. However, the causal link between the intake of a certain substance, such as a vitamin, and the prevention of infectious events is not always demonstrated, so even today, the intake of nutraceuticals to prevent infections, especially respiratory infections, is the subject of debate. This concept becomes even more important in the context of the ongoing COVID-19 pandemic since we are certainly at the center of an epochal historical event that is marking our lives.

In this review, we summarize the evidence supporting a role for nutraceuticals such as probiotics, prebiotics, resveratrol, hesperidin, quercetin, lactoferrin, zinc, Omega-3 (Ω-3) Fatty Acids, and vitamins A, C, D, and K in the prevention of viral infections, especially SARS-CoV-2, in children,.

## 2. Probiotics and Prebiotics

Probiotics are living microorganisms that can have beneficial effects on the host if ingested in a certain quantity. Prebiotics are non-digestible micronutrients, often oligosaccharides, which selectively stimulate the growth and activity of one or a limited number of bacterial species of the intestinal bacterial flora, contributing to the reduction of intestinal pH, thus making the environment inhospitable for pathogenic bacteria. Synbiotics are a mixture of probiotics and prebiotics [26]. The concept of probiotics is closely related to that of microbiota, which is the set of microorganisms that populate a specific organ. It has been widely demonstrated that maintaining an adequate microbiota contributes to the body’s well-being [27,28,29]. The known beneficial effects of probiotics schematically include the following: the biosynthesis of vitamin K; metabolic effects of fermentation of undigested dietary fibers; a positive influence on intestinal peristalsis; and modulation of the immune response. Probiotics seem to have a hitherto undefined role in modulating mucosal immunity. They can regulate the activity of many cells of the immune system, including both innate immunity (NK cells, macrophages, granulocytes, dendritic cells, and epithelial cells) and adaptive immunity (Th1, Th2, Th17, Treg cells, and lymphocytes B) [30]. However, the biomarkers that can explain this mechanism have not yet been identified: some studies have shown an increase in both CD4 + and CD8 + lymphocytes in peripheral blood but have not shown effects on phagocytosis or lymphocyte activation; other studies have shown an increase in blood levels of IFN-γ, but further studies are certainly needed to better define its immunomodulatory effects [27,31]. However, experiments in mice suggest that the intestinal microbiota can stimulate the innate immunity against some respiratory viruses, such as the influenza virus, through the production at the NLRP3 inflammasome level of pro-inflammatory cytokines, such as IL-1 and IL-18, and in turn, can up-regulate the expression of costimulatory molecules (CD80/CD86) on dendritic cells and thus favor the adaptive immune response in the lung [32]. This effect has been hypothesized to depend on a tonic release in the entero-hepatic circulation of microbial ligands for the Pattern Recognition Receptors (PRR) of innate immunity cells. The production of bacteriocins and short-chain fatty acids (especially butyric acid) could also inhibit the replication of pathogens [33]. Probiotics have been used successfully both in the treatment of gastrointestinal forms and in their prevention. Some probiotic strains also seem able to modify inflammatory processes of an allergic nature, according to the observations of studies, even in the medium-term [34,35]. Furthermore, recently, beneficial effects have been highlighted on the use of probiotics, also and especially in immunocompromised patients [36,37,38]. Regarding COVID-19, experience with other viral strains, such as influenza, rhinovirus, and respiratory syncytial virus, has led some to conclude that probiotic supplementation can be considered for the prevention of SARS-CoV-2 infection. However, there is no specific evidence on the subject and specifically in children [39,40,41].

## 3. Resveratrol

Resveratrol (3,4,5-trihydroxy-trans-stilbene) (Figure 1) is a polyphenolic compound present in various plant species, some of which are consumed by humans (mulberry, peanuts, and mainly red grapes/wine), where it performs antifungal functions. This molecule is classified as a phytoalexin, and the main source is *Vitis vinifera*, the common vine, which can contain (at the skin level) from 50 to 100 mg/g. Many biological actions have been attributed to resveratrol, including antioxidant, anti-inflammatory, antiplatelet, cardioprotective, anticarcinogenic, and immunomodulatory activities, as well as induction of lymphocyte proliferation, activation of NK cytotoxicity, and mechanisms of apoptosis regulation [42]. In vitro experiments on animal and human cells have shown that resveratrol has antiviral activity against numerous viruses (VZ, Herpes simplex, polyomavirus, influenza A, HIV) but also parasites (Leishmania) and bacteria (Serratia, Neisseria) [43,44,45,46,47,48,49]. Several recent studies have shown that resveratrol inhibits SARS-CoV in vitro [50,51], and in particular, a study of molecular docking showed that resveratrol revealed a strong interaction SARS-CoV-2 spike protein and human ACE2 receptor complex [52]. Moreover, resveratrol activates Nrf2, which is responsible for adaptation of cells under oxidative stress and inflammation, by decreasing the expression of its negative regulator, KEAP1, and activates SIRT1 deacetylase [53]. When Nrf2 pathway is active, the Keap1-Nrf2 cytoplasmic complex dissociates and Nrf2 migrates to the nucleus stimulating transcription of the target genes with antioxidant response element (ARE) sequences in their promoters [54,55]. Activation of these genes protects cells from oxidative stress and helps them in suppression of inflammation, through the Nrf2 transcriptional repressor activity, inhibiting expression of the inflammatory cytokines IL-1β, IL-6, and TNFα [56,57]. Recently studies showed that oral administration of resveratrol leads to a decrease in the levels of inflammatory cytokines and an activation of the expression of Nrf2 target genes enhancing synthesis of endogenous glutathione and protects alveolar epithelial cells from oxidative stress [58,59]. Finally, resveratrol intake might have a significant effect on susceptibility to or severity of SARS-CoV-2 infection, but the greatest obstacle to this is its poor oral bioavailability, limiting its absorption and, therefore, its therapeutic actions. For this reason, there is still no great evidence supporting its efficacy in the clinical setting [60].

## 4. Hesperidin

Hesperidin is a flavanone glycoside (a subclass of flavonoids) commonly found in citrus fruit, such as sweet oranges and lemons (Figure 1). It is a β-7-rutinoside consisting of an aglycone (Hesperetin) and the disaccharide Rutinose, composed of Glucose and Rhamnose. Also called bioflavonoid because of its various biological properties, hesperidin has been tested for several pharmacological activities, such as anti-atherogenic, antihyperlipidemic, antidiabetic, cardioprotective, antioxidant, and anti-inflammatory actions [60,61]. Before the COVID-19 pandemic outcome and the related in-depth studies, only a few works associated an antiviral activity to hesperidin treatment: authors reported a significant activity against the influenza A virus through a marked reduction of virus replication in two ways [62]. Hesperidin enhanced cell-autonomous immunity via upregulation of p38 and cJun NH(2)-terminal kinase (JNK) expression, which is essential for cell defense mechanisms against influenza virus, selectively modulating the MAP kinase pathways [62]; moreover, hesperidin prevented the influenza A virus replication by inhibition of viral sialidase activity that is involved in the entry and release stages on virus infection [63]. Several recent papers suggested that hesperidin presents a chemical-physical structure suitable for binding to critical proteins involved in the functioning of SARS-CoV-2 [64,65]. Results showed that, among a range of natural substances with potential antiviral effects, hesperidin was the most suited to binding the Angiotensin-converting enzyme 2 (ACE2) interface, highlighting hesperidin’s capability to disrupt the interaction of ACE2 with the receptor-binding domain (RBD). Furthermore, it has been reported that there is a potential interaction between hesperidin and the protease that allows the processing of the first proteins transferred from the viral genome into functional proteins in the host cell, showing that hesperidin has a stronger interaction with the SARS-CoV-2 protease than lopinavir, which is currently in clinical trials for COVID-19 [64,65,66,67]. The cytokine storm, a condition exhibited by patients with COVID-19, is an inflammatory response that evolves into an uncontrolled over-production of soluble markers of inflammation, such as Interferons (IFNγ), interleukins (IL-1β, IL-6), and TNF-α [68]. In this regard, the high anti-inflammatory activity of hesperidin, capable of inhibiting and controlling the release of these inflammatory markers, has been reported [69]. Since multiple studies have confirmed the absence of adverse side effects after oral intake and the overall high safety profile of hesperidin, this flavanone might be useful as a prophylactic, exploiting its marked anticoagulant properties: in fact, the immune response to COVID-19 infection follows the activation of coagulation pathways, promoting clot formation and predisposing patients to the development of multiorgan failure in case of disseminated intravascular coagulation [70]. In this context, the prophylactic administration of hesperidin, combined with the current therapy of diosmin mixture with heparin, could enhance protection against thromboembolism [71,72,73]. A hyperactive inflammatory state caused by the cytokine storm is probably the main cause of pulmonary fibrosis responsible for severe and, in some cases, fatal lung lesions. Pulmonary fibrosis is a pathological consequence of interstitial pulmonary diseases and is characterized by the excessive deposition of collagen and extracellular matrix, as well as the destruction of normal pulmonary architecture [74]. Among the pathological mechanisms involved, oxidative stress, increased production of ROS, and excessive expression of TGF-β, FGF, and PDGF are known to play a crucial role in the disease development [75,76]. A recent study, performed before the COVID-19-era, demonstrated that hesperidin administration ameliorated the altered conditions due to bleomycin-induced pulmonary fibrosis. The treatment enhanced the lung index, the percent oxygen saturation, and the alkaline phosphatase (ALP) and lactate dehydrogenase (LDH) levels. Moreover, hesperidin significantly reversed both the bleomycin-induced down-regulation of Nrf2/HO-1 levels and the up-regulation of TNF-α, IL-1β, IL-6, collagen-1, TGF-β, and Smad-3 mRNA expressions, leading to the amelioration of the oxido-inflammatory condition and the reduction of collagen deposition during pulmonary fibrosis [77].

## 5. Quercetin

Quercetin (3,3′,4′,5,7-pentahydroxyflavone) (Figure 1) is classified as a flavonol, one of the six subcategories of flavonoid compounds, and is the most abundant dietary polyphenolic flavonoid found in apples, berries, brassica vegetables, grapes, onions, tea and tomatoes [78]. As one of the most important plant molecules, quercetin showed several pharmacological activities, such as pro-metabolic, anti-inflammatory, and antiviral effects [79]. Few studies confirmed the quercetin potential as an antiviral agent, highlighting its effect on entry and consequent infection of different influenza viruses, including H1N1, H3N2, and A/WS/33 [80]. Virus entry is the initial step of the viral replication cycle; prevention of this crucial step leads to the suppression of viral infectivity. Quercetin performed the inhibitory activity in the initial stage of influenza virus infection through the interaction with influenza hemagglutinin protein that is responsible for the binding of the virus to host cells [81]. Moreover, quercetin has been implicated in the inhibition of the JNK pathway, essential for the activation of both varicella-zoster-virus (VZV) and human-cytomegalovirus (HCMV) replication [82]. One of the first papers exploring the antiviral effect of flavonoids on coronaviruses showed that quercetin reduced infectivity of human and bovine coronaviruses, OC43 and NCDCV [83], while following experiments demonstrated the capacity of quercetin to antagonize HIV-luc/SARS pseudotyped virus entry [84]. Although recent studies suggested the role of quercetin as a possible candidate to control the replication and the early phases of SARS-CoV-2 infection [85,86,87,88], the poor pharmacokinetic profile of this flavonoid, because of its low oral bioavailability, requires the use of a galenic formulation aimed to improve the rate of absorption that ranges from 3% to 17% in healthy individuals receiving 100 mg [79]. Only a few clinical trials have been conducted to analyze the effect of quercetin on the prophylaxis and treatment of COVID-19. On the one hand, the assumption of quercetin as a strong scavenger and anti-inflammatory agent, had effective results in COVID-19 cases, with the scheduled dosage of 500 and 1000 mg for prophylaxis and treatment, respectively [89]. On the other hand, recruitment of three different clinical trials with low risk of bias showed that only one case reported a decrease in the incidence of upper respiratory tract infections following an athletic event. Indeed, although quercetin exhibited both immunomodulatory and antimicrobial effects in preclinical studies, benefits were only obtained in older, athletic adults [90].

## 6. Lactoferrin

Lactoferrin (fragment involved in iron binding shown in Figure 1) is a basic glycoprotein; it belongs to the transferrin family and comprises 692 amino acids. Lactoferrin is found in breast and bovine milk, and is particularly concentrated in colostrum, the first milk produced in the days immediately following birth. Lactoferrin is a glycoprotein capable of subtracting unbound iron from body fluids and areas of inflammation with twice the capacity of transferrin to avoid the damage produced by toxic oxygen radicals and decrease the availability of ions ferric for the microorganisms that invade the host [91]. Lactoferrin has antibacterial, antiviral, antioxidant, and immunomodulatory functions. As for viral infections, lactoferrin appears to inhibit the attack of viruses on their receptors on human cells [92]. The potential spectrum of activity of lactoferrin against SARS-CoV-2 comes from observations on SARS-CoV [93]. It has been shown that lactoferrin has antiviral properties mainly through three mechanisms: direct binding to the virus; binding of lactoferrin with heparan sulfate proteoglycans (HSPGs), which are the adhesion molecules of many viruses on the surface of the host cell, thus acting with a competitive mechanism; and intracellular inhibition of viral replication [92,93,94]. It is also able to enhance the antiviral action of T and NK 20 lymphocytes [91]. Based on these observations, a clinical study was proposed for paucisymptomatic and asymptomatic COVID-19 patients to evaluate the efficacy and safety of an innovative liposomal formulation of lactoferrin administered for oral and intranasal use [95].

## 7. Vitamin C

Vitamin C (L-Ascorbic acid, AA) (Figure 1) belongs to water-soluble vitamins and is involved in many different biochemical mechanisms related to the cellular environment of most living organisms [96,97]. Among them, humans are not able to synthesize AA from D-glucose due to the lack of L-gluconolactone oxidase, one of the enzymes involved in its biosynthetic pathway [96]; because of that, its intake through diet is essential to satisfy the physiological requirements [97]. AA represents one of the most known “scavenger molecules”, showing an efficient antioxidant activity [98,99,100]. In fact, AA scavenges free radicals and ROS, products of physiological cell metabolism or associated with inflammatory diseases, and oxidative damage [101]. In addition to its antioxidant properties, vitamin C is involved in the regulation of many biological pathways behaving as a multifunctional cofactor [102]. AA plays a pivotal role in a plethora of significant mechanisms, such as biosynthesis of corticosteroids, catecholamines, collagen, and carnitine, as well as tyrosine oxidation and epigenetic modifications [102,103]. Moreover, many studies have identified AA as an immune-modulator affecting the proper function of several cell types included in innate and acquired immunity [104,105,106,107,108,109]. The clinical and biological importance of vitamin C is further confirmed by many studies reporting its adjuvant properties towards viral infections, such as herpes virus (HSV), influenza type 1, HIV, and rhinovirus [110,111,112,113], but also sepsis and inflammation [114,115,116,117]. In the context of the ongoing COVID-19 pandemic and while there is still no effective antiviral therapy, this antiviral hallmark has a new significance. In this regard, many authors have reported a strong affinity between the beneficial properties of AA towards the complex pathological onset due to SARS-CoV-2 infection [118,119,120,121,122]. In this context, vitamin C seems to act in several ways as a polyfunctional molecule. Different studies show that AA’s antioxidant property is useful to reduce the oxidative damage in infected cells inducing inflammatory lung injury, often associated with mechanical ventilation [123,124,125,126,127,128]. In the pathological context of a virus outbreak, cytokines represent a crucial system of signaling involved in phlogosis in response to infection [129]. In particular, in vitro studies show that vitamin C can decrease the release of inflammatory cytokines IFN-γ, IL-6, and TNF-α, both in a model of inflammation with peripheral blood lymphocytes treated with Lipopolysaccharides (LPS) [108] and in monocytes isolated from pneumonia patients [130]. The altered redox homeostasis and the oxidative environment contribute to activating cell pathways exploited by the virus to ensure its survival and to suppress the host immune response. In this regard, vitamin C seems to be essential to enhance the immune system through the improvement of interferon synthesis [115] but also through the management of immune cellular subtypes [102]. Particularly, vitamin C affects the correct functionality of leukocytes, neutrophils, and lymphocytes, in addition to modulating many inflammatory mediators [102]. Neutrophils and monocytes accumulate AA, useful to the stabilization of cellular microtubules, which are essential for chemotaxis and cellular migration [131,132,133]. Moreover, vitamin C supports neutrophils to phagocyte and kill pathogens [134], manages the following caspase-dependent apoptosis, and enhances the neutrophils clearance by macrophages attenuating phlogosis in situ [102,135]. Acquired immunity represents another important target which function is strictly associated with AA: in this regard, B- and T-Lymphocytes accumulate vitamin C [136] which seems to be involved in cell proliferation, inducing an increased antibodies production resulting in enhanced cell survival [137,138,139]. The various roles of vitamin C and its involvement in COVID-19 clinical implications are shown by many clinical studies that confirm the efficiency of AA supplementation towards symptoms, duration of intensive care unit (ICU), and clinical resolution of the disease [119,120,128,140,141].

## 8. Vitamin D

Vitamin D (Figure 1) can be defined as a hormone in all respects. In the last decade, extraskeletal effects of vitamin D have been discovered which are not directly related to calcium metabolism [142]. It has been documented that, among its many actions, vitamin D enhances the body’s immune response: children exposed to recurrent diseases of the airways often have sub-optimal levels of vitamin D (measurable in the blood) and that the correction of the deficit results in an improvement in the state of health and a reduction in the number of infectious episodes [142]. This vitamin exerts an anti-infective activity through its ability to favor the synthesis of some peptides with powerful antimicrobial activity (cathelicidin or LL37 protein). It has also been shown that vitamin D intervenes directly in the signaling mechanisms downstream of the T-cell-receptor (TCR) of the T cell, in the maturation of dendritic cells, in the polarization of immune responses, in the promotion of Treg responses, in the production of IgE, and in the recruitment of eosinophils in the airways [142,143,144,145]. In addition, several human diseases have been associated to vitamin D deficiency and consequently dysregulation in both ROS and Ca2+ signaling. Vitamin D treatment resulted in an increasing of Nrf2 expression that also control Ca2+ and redox signaling [146]. Furthermore, Tao et al. have shown that vitamin D protected against lung injury through induction of autophagy in a Nrf2-dependent manner, decreasing Nrf2 ubiquitination and increasing its protein stability [147]. Several studies have shown that vitamin D supplementation can help improve lung function in children with bronchiolitis and reduce the likelihood of developing respiratory infections [148,149,150]. Based on these observations, McCartney and Byrne recently hypothesized that daily administration of vitamin D to obese individuals, healthcare professionals, and smokers might improve resistance to COVID-19 [151]. Grant et al. pointed out that through several mechanisms, vitamin D can reduce the risk of viral infections and that vitamin D deficiency can contribute to acute respiratory distress syndrome. These authors suggest that for the prevention of infection and spread of COVID-19, you should use 10,000 International Units (IU) per day of vitamin D3 for a few weeks to rapidly increase concentrations of 25-hydroxy-vitamin D levels, followed by 5000 IU/day. They indicate that for the treatment of COVID-19, higher doses of vitamin D3 should be used [152]. However, it is worth mentioning that most clinical trials have not reported a significant reduction in the proliferation of SARS-CoV-2 with vitamin D supplementation. Greiller et al. point out that in the in vitro culture of human respiratory epithelial cells, vitamin D modulated the expression and secretion of interferon type 1, chemokines (including CXCL8 and CXCL10) and pro-inflammatory cytokines (such as TNF and IL-6). However, they suggest the need for further studies that might clarify the effects of vitamin D metabolites against viral replication [153]. According to Wimalawansa, patients who have micronutrient deficiency, especially those with hypovitaminosis D (given the reduction in the production of anti-inflammatory cytokines), have the greatest risk of developing viral diseases. Therefore, these authors recommend the use of vitamin D supplements and/or exposure to the summer sun to increase serum concentrations of 25-hydroxy-vitamin D above 30 ng/mL to strengthen the immune system [154]. Wimalawansa also points out that to experience adverse effects with the use of vitamin D would require doses above 25,000 IU for many months, or to take 1 million IU daily for a few days [154]. Gasmi et al. reported that vitamin D supplementation is effective when used before the onset of the respiratory tract infection [155]. However, for the treatment of patients at risk of COVID-19, it is recommended to increase the concentrations of 25-hydroxy-vitamin D to between 40–60 ng/mL (100–150 nmol/L), considering the use of 10,000 IU per day of vitamin D3 for a few weeks with the aim of rapidly increasing concentrations, followed by 5000 IU/day [155]. A large cross-sectional study sought to associate data on morbidity and mortality caused by COVID-19 and average serum vitamin D levels in European countries and showed a strong association between low vitamin levels and cases of infection by coronavirus for every million inhabitants [156]. As a limitation, the authors reported that the cases in each country could be influenced by several factors, such as the number of tests performed [157]. Furthermore, another retrospective observational study analyzed vitamin D concentrations in 107 patients, finding significantly lower levels in patients positive for COVID-19 than negative patients [156]. These authors raise the hypothesis that supplementation of this vitamin could reduce the risk of contracting COVID-19 [156]. Studies also sought to assess whether good levels of vitamin D would be able to assist in the treatment or improve the prognosis of patients with COVID-19. As a retrospective observational study, in which the authors sought mortality data from COVID-19 and compared these to vitamin D levels in different countries in Europe, it was found that in countries with a lower latitude and with high rates of vitamin D deficiency, the highest rates of both infection and death occurred on the European continent [158]. Through a randomized, double-blind clinical trial, there was an assessment of the effectiveness of treating patients hospitalized with COVID-19 with vitamin D; 50 patients received vitamin D, and 26 were in the control group [159]. Vitamin D administration significantly reduced the need for ICU treatment of patients requiring hospitalization due to proven COVID-19. However, serum vitamin D levels were not measured before, during, or after the experiment [159]. An observational study, which presented a protocol for nutritional supplementation early on to patients hospitalized with COVID-19, reported that this protocol would be important since many patients acquired an inflammatory condition, in addition to anorexia and difficulties in eating food, leading to an increase in the percentage of respiratory failure [160]. According to the protocol, if vitamin D deficiency was found, supplementation would be recommended to patients due to the evidence that, if present at normal levels, vitamin D could improve immunity during treatment, in addition to reducing inflammation [160]. On the other hand, the authors do not present evidence of the effectiveness of using this protocol as a form of treatment for COVID-19. A retrospective observational study, using data from medical records and aiming to identify which factors were associated with hospitalization and severity of patients with COVID-19, pointed to vitamin D deficiency together with other factors such as diabetes, high cholesterol, and asthma, in addition to kidney and cardiovascular diseases, as factors with a greater chance of hospitalization [161]. However, the authors report that this represents the population as a whole because their sample consisted of only one health system. However, not all studies have found associations between vitamin D and COVID-19. A large study in the UK, investigating whether there is an association between vitamin D levels and COVID-19 through univariate analysis, found no significant relationships in a sample of 449 infected patients [162]. More studies are needed to clarify the interactions between vitamin D and COVID-19. Several types of research are being carried out internationally, aiming to investigate whether different outcomes are modified with vitamin supplementation. The main variables being researched are related to decreasing the incidence and time of infection, the length of hospital stay, admission to the intensive care unit, and decreasing the incubation period and mortality. Awaiting the disclosure of these results from studies with great quality and methodological rigor, the benefit of Vitamin D for the treatment or prevention of patients with COVID-19 may be better estimated. Recent meta-analyses have tried to draw conclusions on the role of vitamin D in COVID-19. Liu et al. in a meta-analysis involving 361,934 participants found that low vitamin D levels are associated with an increased risk of SARS-CoV2 infections [163]. Beyond the risk of infection, the conclusive data that emerge on severity of COVID-19, risk of hospitalization, mechanical ventilation, and mortality seem more interesting. Yisak et al. showed that serum vitamin D levels can influence not only the chances of being infected with SARS-CoV2 but also the seriousness of COVID-19 and mortality from COVID-19 [164]. However, the meta-analysis by Pereira et al. seems to highlight that vitamin D deficiency would not be associated with a higher risk of SARS-CoV2 infection (OR = 1.35; 95% CI = 0.80–1.88) but with a more severe COVID-19 (OR = 1.64; 95% CI = 1.30–2.09) in terms of increased hospitalization (OR = 1.81, 95% CI = 1.41–2.21) and mortality (OR = 1.82, 95% CI = 1.06–2.58) [165]. More complex is the debate on the blood concentration of vitamin D that should be considered protective. Pereira et al. define the state of deficiency with consequent increased risk of serious manifestations when the levels are below 50 nmol/l [165]. Jain et al. compared the levels of vitamin D between asymptomatic and symptomatic patients, showing that the former had mean serum levels of vitamin D equal to 27.89 ± 6.21 ng/mL compared to 14.35 ± 5.79 ng/mL in the latter [166]. In this sense, it is interesting to note that the Mediterranean diet positively influences serum levels of vitamin D, and this could be an advantage; however, further studies are needed to support this hypothesis [167,168]. 

## 9. Zinc

Zinc is an essential mineral present in the body in quantities greater than that of any other trace element other than iron. Zinc performs various functions: it is essential for the functioning of many enzymes; it is necessary for the functioning of some cellular mediators; it contributes to the stabilization of the cell membrane (cytoskeleton); and regulates apoptosis by lymphocytes in vitro and in vivo [169]. The role of zinc in the integrity of the immune system is well known, and health interventions, such as zinc supplementation, have been hypothesized to prevent alteration of the immune system and improve resistance to infections [170]. Zinc is considered useful for the respiratory mucosa as it seems to increase the beat frequency of cilia, resulting in an improvement in mucociliary clearance; moreover, this element contributes to inhibition of the replication of some viruses, such as influenza and rhinoviruses, and for this reason, it might also be effective in inhibiting the replication of SARS-CoV-2 [171,172,173,174,175,176,177]. Regarding the association with COVID-19, the role of zinc was analyzed for its synergistic action with chloroquine and hydroxychloroquine, drugs to which antiviral properties had been attributed [178,179]. However, specific data on children are lacking. 

## 10. Vitamin A

Vitamin A1 or Retinol is a fat-soluble vitamin and an essential dietary factor because it is not synthesized de novo by the human body. The main sources of this vitamin are organ meats, milk, and cheese. Although provitamin A carotenoid is present in green vegetables and yellow fruits, carotenoids need to be cleaved to retinal before absorption [180]. In particular, preformed vitamin A (Figure 1)— also known as retinol, retinal, retinoic acid, and retinyl ester—is hydrolyzed into retinol in the lumen of the small intestine. During a deficiency status, vitamin A is mobilized, and retinol can circulate bound to the retinol-binding protein (RBP) utilized by target tissues [180]. Vitamin A functions are mediated by all-trans-retinoic acid, which regulates the expression of several genes by binding specific nuclear transcription factors [180,181,182,183]. These genes are involved in crucial biological activities supporting vision, growth, and cell and tissue differentiation, as well as hematopoiesis and immunity. In immunity, vitamin A supports the integrity of gastrointestinal epithelial tissue among children suffering from severe infections or who are undernourished [180]. This vitamin is also important in the regulation of NK cells, macrophages, and neutrophils [181]. In particular, vitamin A plays a regulatory role in the early differentiation stage of NK cells, causing downregulation of IFN-γ and upregulation of IL-5. Furthermore, it regulates the differentiation of dendritic cell precursors and promotes the secretion of pro-inflammatory cytokines IL-12 and IL-23. Vitamin A also exhibits a fundamental role in promoting Foxp3+ Treg generation, inhibiting Th1/Th17 generation and the Th9 transcriptional program [184,185]. Adaptive immunity also involves vitamin A; certainly, retinoids are physiological modulators of normal B cell growth and differentiation, and a vitamin A deficiency affects B cell function [182]. This involvement is supported by animal studies which showed impairment in the antibody response due to vitamin A deficiency [183]. Antibody production could be enhanced by the action of vitamin A on T helper 2 cells development and antigen-presenting cells (APCs) [180,186,187]. Moreover, retinoids play a role in cell-mediated immunity, representing an important cofactor in T cell activation and acting on the expression of membrane receptors that mediate T-cell signaling [188]. Vitamin A supplementation in pediatric populations shows the potential effect to increase T-cell and, in particular, the CD4 subpopulation [180,189]. The recommended daily dosage for infants up to 12 months of age is 450 μg. The dietary reference intakes in older children differ based on the age-sex group: from 400 μg to 700 μg (female) or 900 μg (male). A weekly dose of Vitamin A at the RDA level reduced the rate of mobility and mortality in children at risk of a vitamin A deficiency, probably associated with a viral infection such as measles [190]. A meta-analysis of 47 studies—including 1,233,856 children—defined a reduction of 12% of all-cause mortality during vitamin A supplementation [191]. This association could find an explanation by the low-to-moderate evidence that vitamin A supplementation in children can reduce the incidence of diarrhea and measles [192,193]. In contrast with the cited trial [191], in a large cluster-randomized trial (DEVTA trial),which included more than 1 million pre-school children in North India, the supplementation with high-dose vitamin A (200,000 UI every six months) did not achieve a significant reduction in mortality. Nonetheless, the same article highlights a meta-analysis (that includes DEVTA and eight other previous randomized trials) reporting a weighted average mortality reduction of 11% [190]. There are also several studies in children regarding the role of vitamin A in respiratory syncytial virus infection treatment, which showed no effect of vitamin A in the reduction of incidence of lower respiratory tract infections [194,195,196,197]. It is also necessary to take into consideration the possible hepatotoxicity of Vitamin A supplementation. As demonstrated, hypervitaminosis A could lead to liver injury comprises mild elevations of serum liver enzymes, cholestatic hepatitis, non-cirrhotic portal hypertension, and progressive fibrosis and cirrhosis [198,199,200,201]. However, toxicity does not usually occur with standard doses below 50 000 IU/day, but individual tolerability may vary [202]. However, taking into consideration the role of vitamin A in the regulation of the immune system, supplementation should be offered to children at risk of deficiency.

## 11. Omega-3 (Ω-3) Fatty Acids

Ω-3 fatty acids (Figure 1) are a type of polyunsaturated fatty acids (PUFAs) characterized by the presence of a double bond at the Ω-3 carbon atom. Among these, the simplest is α-linolenic acid (18:3n-3), which is synthesized from the Ω-6 linolenic acid (18:2n-6) through desaturation and catalyzed by Δ-15 desaturase. The two cited acids are essential fatty acids (EFAs) and cannot be synthesized sufficiently by humans, so they must be obtained from the diet [203]. However, animals can metabolize α-linolenic acid by further desaturation and elongation to obtain eicosapentaenoic acid (known as EPA) and docosahexaenoic acid (known as DHA). The same enzymes are employed by Ω-6 fatty acids for the metabolic pathways, leading to the production of arachidonic acid: this means that the α-linolenic acid is a competitive inhibitor of linoleic acid metabolism and vice versa [204]. However, the conversion to EPA and DHA is poor in humans–with reported rates of less than 15%–and these fatty acids must be supplied with food: α-linolenic acid is present in plant oils while DHA and EPA are present in fish, fish oils, and krill oils [205]. Ω-3 fatty acids are a fundamental component of the phospholipids that form the structure of cell membranes. Moreover, they provide energy for the body operating several functions in the cardiovascular, pulmonary, immune, and endocrine systems [159]. 

Indeed, both Ω-3 and Ω-6 metabolites play a role in immunity regulation [206]. PUFAs serve as a substrate for the enzymatic production of molecules involved in the resolution of inflammation, defined specialized pro-resolving mediators (SPMs) [207,208]. These molecules are different from the immunosuppressive agents because they further display antimicrobial action promoting host defense [209]. These SPMs are classified as resolvins, protectins, and maresins: these molecules are important in supporting immune cell function, neutralizing and eliminating pathogens with the resolution of inflammation [210]. Resolvins, in particular, are effective in the inhibition of neutrophil migration, reducing the entry of further neutrophil in the inflammation site [159,211]. Moreover, SPMs has a strong anti-inflammatory action, reducing neutrophil activation and preventing tissue damage [210,212]. They are also able to stimulate NK cells to trigger granulocytes apoptosis, accelerating the clearance of apoptotic polymorphonuclear leukocytes [203]. It has been demonstrated that pro-resolving mediators up-regulate CCR5 expression on apoptotic activated T cells, causing the sequestration of pro-inflammatory cytokines and leading to the resolution of the inflammation [213]. Several studies have investigated the link between Ω-3 fatty acid supplementation and respiratory infections/illness and–in particular–the role in improving the ARDS [4,214,215,216,217,218,219,220,221,222,223,224,225,226]. Some authors studied the effects of Ω-3 fatty acid supplementation on infant morbidity caused by respiratory infections, wheezing, and asthma. A randomized controlled trial, which included 736 pregnant women and a total of 695 children, showed a reduction of the risk of persistent wheeze or asthma (approximately 7%) in the first 5 years of life among children of women who received daily supplementation with Ω-3 PUDA (EPA/DHA) during the third trimester of pregnancy [217]. Furthermore, supplementation was also associated with a reduced risk of infections of the lower respiratory tract [217]. Severe COVID-19 could manifest as a hyperinflammatory syndrome, which is characterized by an important hypercytokinaemia (cytokine storm) with multiorgan failure and ARDS in approximately 50% of patients [4]. Several studies have been done to determine if Ω-3 fatty acids DHA and EPA could modulate the systemic inflammatory response, affecting plasma cytokine production [213,220,223]. In a randomized controlled trial, the researchers demonstrated the association between the consumption of Ω-3 PUFAs and fewer episodes and shorter duration of illness in the upper respiratory tract: in particular, there was a significantly lower concentration of TGF-β1 concentration compared with the placebo group [220]. A recent systematic review reported a significant improvement in blood oxygenation and in the duration of ventilator days and ICU length of stay in patients with ARDS who received nutrition containing antioxidants and rich in EPA and DHA, although there was a low quality of evidence [223]. Animal studies–supporting these findings–might suggest a potential role for EPA and DHA in reducing lung injury with the support in the resolution of inflammation, through the production of SPMs [213]. Nonetheless, further trials and studies are necessary to support this hypothesis.

## 12. Vitamin K

Vitamin K (Figure 1) belongs to the fat-soluble vitamin family and represents a fundamental nutrient involved in many biochemical and physiological processes. Among them, vitamin K is involved in the coagulation pathway, acting as a cofactor for many proteins, including prothrombin, factors VII, IX and X, protein C, and protein S (PROS1) [227]. Vitamin K exists in two different chemical and biological subtypes: vitamin K1 (phylloquinone-PK) and vitamin K2 (menaquinone-MK). Although these two different forms are chemically and functionally distinct, they share many biological functions, like the γ-carboxylation catalysis of vitamin K-dependent proteins (Gla proteins) [228], a post-translational modification of glutamate through which vitamin K is involved in most of the biological mechanisms, including coagulation, calcium homeostasis, bone and vascular mineralization [229,230]. Besides coagulation, vitamin K is involved in the immune response associated with vascular damage: it enables the interaction between TAM receptors (a subfamily of tyrosine kinases receptors) and their Gla-proteins ligands, PROS1 and growth-arrest-specific 6 (GAS-6) [231,232]. Based on this evidence, several studies hypothesize an interesting link between vitamin K and COVID-19 clinical outcome and related complications, including thromboembolism and coagulopathy [228,229]. In fact, the clinical course of the infection is commonly associated with vascular calcification and stiffening of lung tissue; this is obviously due to elastic fibers degradation, which mainly affects the lung and the arteries [229]. Interestingly, vitamin K2 is involved in the activation of matrix Gla-protein, which represents the main calcification inhibitors in soft tissues [228]. This evidence suggests a correlation between vitamin K deficiency in COVID patients and worsens clinical course due to the enhanced fiber mineralization. Actually, it has been reported that vitamin K deficiency is strongly associated with admission to ICU [233]. In addition, the low levels of vitamin K seem to contribute to increasing levels of IL-6 release and Th2 storm activation [234].

## 13. Conclusions

This study is a review of the literature that focuses on nutraceuticals used or studied in the prevention of infections in children. Clinical evidence has shown that numerous therapies based on natural products or food supplements with immune-stimulatory and antiviral actions capable of supporting and increasing the body’s immune defenses might be able to reduce the duration and severity of symptoms related to colds, flu, and respiratory viruses in general, as well as to prevent the onset of serious complications. The main antiviral actions and eventual clinical evidence of the nutraceuticals addressed in this review are summarized in Table 1 and in Figure 2. However, many of these highlights derive from in vitro experiments since it is difficult to propose randomized clinical trials for substances such as nutraceuticals. As a result, there is a lack of strong scientific evidence. However, there is reassuring data concern the safety of these molecules, as in the vast majority of studies and with most of the products, no significant secondary effects were recorded such as to absolutely contraindicate their use in the common clinical settings in which they are employed. For this reason, their possible use as an adjunct in the prevention or treatment of asymptomatic forms of COVID-19 has been hypothesized, although there is no clinical evidence to document their effectiveness. In addition to the nutraceuticals analyzed, the role of other nutraceuticals has been explored. Curcumin is a yellow polyphenolic curcuminoid from Curcuma longa (turmeric) that seems to have a broad spectrum of bioactivities, including antioxidant, antiapoptotic, and antifibrotic properties with inhibitory effects on Toll-like receptors, NF-κB, inflammatory cytokines and chemokines, and bradykinin [235,236]. Interestingly, Curcumin can be appraised to hinder cellular entry and replication of SARS-CoV-2 and to prevent and repair COVID-19-associated damage of pneumocytes, renal cells, cardiomyocytes, and hematopoietic stem cells [237]. Bromelain is a cysteine protease, isolated from the pineapplestem (Ananas comosus) and is classically used in trauma for its known anti-inflammatory and anti-edema properties [238]. Moreover, bromelain inhibits cyclooxygenase and modulates prostaglandins and thromboxane, affecting both inflammation and coagulation, and also hydrolyzes bradykinin, supporting its potential role in alleviating COVID-19 symptoms [239].

The data collected in our review led to the conclusion that the evidence of the efficacy of nutraceuticals in children is absolutely heterogeneous, and for some of these products, there is no information in the literature to recommend their use. Therefore, it is desirable that there be further independent, rigorously randomized, and controlled studies that take into consideration the clinical effects, rather than the purely biological ones, in vivo or in vitro, of many molecules, both in therapy and in the prevention of respiratory infections. Even less is known about the possible therapeutic applications of these substances during the COVID-19 pandemic for preventive purposes or for the treatment of paucisymptomatic forms, given that specific evidence on the subject is lacking. However, it could be speculated that, given the relative harmlessness of these substances, their use could still be suggested, especially in children.

## Figures and Tables

**Figure 1 ijms-22-02465-f001:**
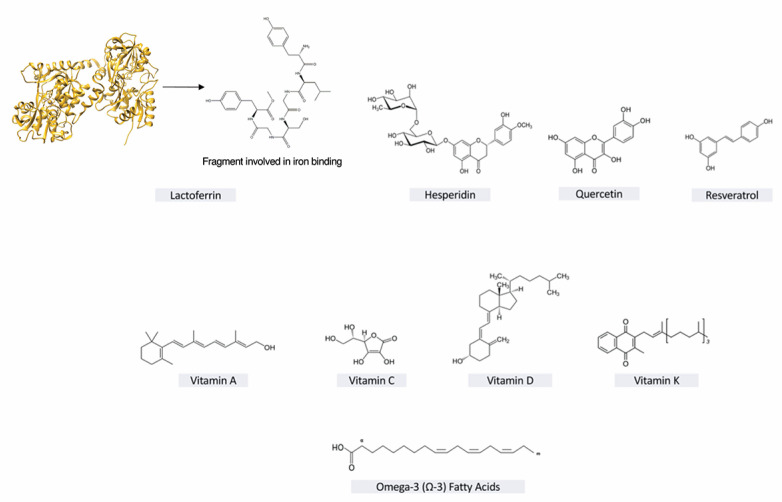
Nutraceuticals’ chemical structures.

**Figure 2 ijms-22-02465-f002:**
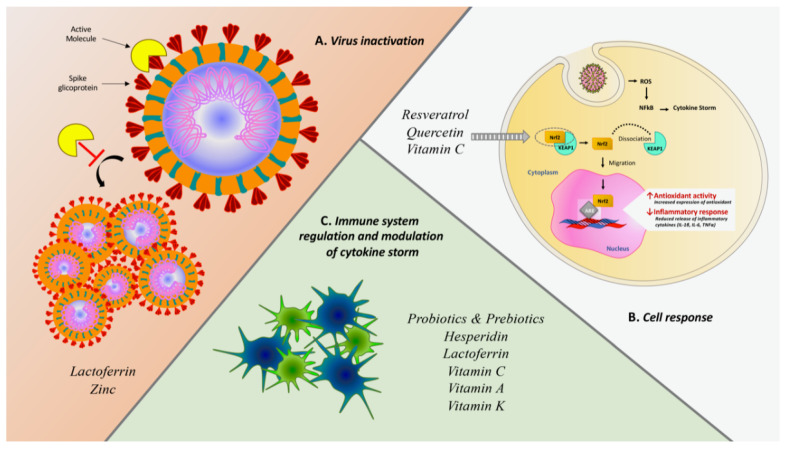
Summary scheme of several mechanisms exploited by nutraceuticals against SARS-CoV-2 infection. (**A**) Antiviral effect due to the direct interaction with SARS-CoV-2 spike protein or inhibitory effect of viral proliferation. (**B**) Cell response induced by the activation of Nrf2 and following genetic regulation downstream. Activated Nrf2 dissociates from its complex with KEAP1 and migrates from cytoplasm to nucleus within which it interacts with ARE (Antioxidant Response Element) sequences producing gene regulation both promoting antioxidant response and reducing the inflammatory cascade (49). (**C**) Indirect immune system modulation through enhancement of immune performance and concurrent decrease of pro-phlogistic cytokines like IL-6 IL-1β and TNFα in order to mitigate the harmful effects of cytokine storm.

**Table 1 ijms-22-02465-t001:** Nutraceuticals, relative antiviral properties, and eventual clinical evidence on SARS-CoV2.

Nutraceutical	In Vitro Effects	Main Viral Targets	Clinical Evidences Toward SARS-CoV2	Type of Study	Participants	Main Findings	Limitations
Probiotics and prebiotics	- Modulation of the innate (NK cells, macrophages, granulocytes, dendritic cells, and epithelial cells) and adaptive (Th1, Th2, Th17, Treg cells, and lymphocytes B) immune systems - Production of bacteriocins and short-chain fatty acids	Influenza viruses, Rhinovirus, Respiratory syncytial virus (RSV)	–		–		
Resveratrol	- Modulation of the immune e antioxidant systems (NF-kB and Nrf2) - Inhibition of viral replication in vitro - Interaction with spike protein and human ACE2 receptor complex	VZV, Herpes simplex (HSV), Poliovirus, Influenza A, HIV	–		–		
Hesperidin	- Enhancement of cell-autonomous immunity (p38 and JNK expression) - Inhibition of the release of pro-inflammatory cytokines - Interaction with ACE2 interface - Interaction with viral proteases involved in the processing of viral proteins in the host cell	Influenza viruses	Prophylactic administration due to the protective effect toward thromboembolism and fibrosis [70,77].	In vitro/In vivo	–		
Quercetin	- Inhibition of JNK pathway- Antagonized HIV-luc/SARS pseudotyped virus entry	Influenza viruses (H1N1, H3N2, A/WS/33), VZV, Cytomegalovirus (CMV), OC43 and NCDCV	Clinical efficacy on prophylaxis and treatment of COVID-19 cases	Clinical Trial	447	Effective dosage of 500 and 1000 mg for prophylaxis and treatment, respectively	Low number of patients
Lactoferrin	- Enhancement of T and NK lymphocyte activity - Inhibition of SARS CoV2 entry and adhesion -Intracellular inhibition of replication - Spike protein block by ACE-2 independent pathway	CMV, HSV, HIV, HCV, HBV, HPV, Rotavirus, Poliovirus, RSV	–	In vitro/In vivo			
				Many clinical studies [95]			
Vitamin C	- Scavenger of free radicals and ROS - Modulation of immune system (decrease the release of inflammatory cytokines IFN-γ, IL-6 and TNFα) - Support of neutrophils-mediated kill pathogens phagocytosis	HSV, Influenza type 1, HIV, Rhinovirus	Clinical efficacy on prophylaxis and treatment of COVID-19	Multicenter, prospective randomized, placebo-controlled trial [240]	308 adult patients into ICU (Wuhan)	New potential therapy for COVID-19 by clarifying the effect of High dose of Intravenous Vitamin C on the prognosis of patients, especially on respiratory function Death or persistent organ dysfunction	Low number of recruited patients No effective standardized guideline for COVID-19 treatment at the early stage
				Many ongoing clinical studies [119]			
Vitamin D	- Modulation of the immune and antioxidant systems - Stimulation of the synthesis of antimicrobial proteins (cathelicidins and LL37)	Respiratory viruses	Clinical data about adjuvant activity in prophylaxis or treatment of COVID-19	Retrospective observational study [157]	European population	Negative correlations between mean levels of vitamin D in each country and the number of COVID-19 cases	Number of tests performed is different among countries
				Retrospective observational study [156]	107 adult patients	Lower vitamin D levels in SARS-CoV2 infected	Low number of patients from a single hospital
				Pilot randomized clinical study [159]	76 adult patients	Administration of a high dose of Calcifediol reduced the need for ICU treatment of patients requiring hospitalization	Serum vitamin D levels were not measured before, during, or after the experiment
				Retrospective observational study [164]	689 patients	Vitamin D deficiency increases chance of hospitalization	Sample consisted of only one health system
				Retrospective observational study [162]	348,598 UK Biobank participants	No significant relationships between vitamin D levels in a sample of 449 SARS-CoV2 infected patients	Blood samples were collected from 2006 to 2010
				Metanalysis [163]	361,934 participants	Low vitamin D status might be associated with an increased risk of COVID-19 infection	No evaluation of clinical severity or prognosis
				Metanalysis [165]	372,332 participants	Vitamin D deficiency is not associated with a higher risk of SARS-CoV2 infection but with a more severe COVID-19	No stratification according to the sex of the participants
Zinc	- Improvement of mucociliary clearance	Influenza, Rhinovirus	–		–		
Vitamin A	- Regulation of NK cells, macrophages, and neutrophils - Downregulation of IFNγ and upregulation of IL-5- Differentiation of dendritic cells’ precursors	Respiratory viruses	–		–		
Omega-3 (Ω-3) Fatty Acids	- Production of pro-resolving mediators (resolvins, protectins, and maresins)	Respiratory viruses	–		–		
Vitamin K	- Modulation of the immune response associated to vascular damage	–	–		–		

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
