# Peer review of "Nutraceuticals in the Prevention of Viral Infections, including COVID-19, among the Pediatric Population: A Review of the Literature"

_ijms, 2021, doi:10.3390/ijms22052465_

Round 1
Reviewer 1 Report
This is a very interesting and well-written review. Attention to the following issues would further strengthen the present manuscript:
- Line 52, second page. Please add have ALSO been reported
- Regarding probiotics, please add some important drawbacks particularly in immunocompromised individuals ( PMID: 32472285, PMID: 31729330, PMID: 33005864, etc)
- A table is necessary presenting clinical studies or meta-analyses (study, number of participants, type of study, effects, limitations) related to nutraceuticals and COVID-19 particularly related to vitamin D, C, A, Omega-3 (Ω-3) Fatty Acids etc.
- Also for the readership, it would be imperative to modify Table #1 with all these nutraceuticals in a column, in vitro effects regarding other viruses, in vitro effects regarding SARS-CoV-2, and in vivo effects. As it is, Table 1 is not specific and not helpful.
- Please analyze in depth after presenting Table (issue#3) the effect of vitamin D on the risk of infection, severity of COVID-19, risk of hospitalization, mechanical ventilation and mortality. There are a couple of meta-analyses dealing with this subject. After that, discuss the effect of supplementation with vitamin D. What are the proposed concentrations of vitamin D that are considered optimal for prevention of severe COVID-19? Please discuss also the role of the Mediterranean diet in vitamin D levels (PMID: 32429342; PMID: 33509003). This is an important issue relating anti-inflammatory diet and vitamin D levels.
- Regarding zinc, please refer also to studies using zinc together with hydroxychloroquine in COVID-19.
- Regarding vitamin A, please discuss the potential of hepatotoxicity with its administration.
- It is important to include immune-boosting nutraceuticals such as curcumin, bromelain (PMID: 33291560, PMID: 33205039, PMID: 33133525)
Author Response
Reviewer 1
This is a very interesting and well-written review. Attention to the following issues would further strengthen the present manuscript:
Line 52, second page. Please add have ALSO been reported
Answer: Thank you. We added “also”.
Regarding probiotics, please add some important drawbacks particularly in immunocompromised individuals ( PMID: 32472285, PMID: 31729330, PMID: 33005864, etc)
Answer: Thank you for your suggestion. We added some drawbacks about immunocompromised individuals.
A table is necessary presenting clinical studies or meta-analyses (study, number of participants, type of study, effects, limitations) related to nutraceuticals and COVID-19 particularly related to vitamin D, C, A, Omega-3 (Ω-3) Fatty Acids etc.
Answer: Thank you for your suggestion. As you requested, we modified and implemented table 1 with your required info.
Also for the readership, it would be imperative to modify Table #1 with all these nutraceuticals in a column, in vitro effects regarding other viruses, in vitro effects regarding SARS-CoV-2, and in vivo effects. As it is, Table 1 is not specific and not helpful.
Answer: Thank you. As you suggested, we modified table 1.
Please analyze in depth after presenting Table (issue#3) the effect of vitamin D on the risk of infection, severity of COVID-19, risk of hospitalization, mechanical ventilation and mortality. There are a couple of meta-analyses dealing with this subject. After that, discuss the effect of supplementation with vitamin D. What are the proposed concentrations of vitamin D that are considered optimal for prevention of severe COVID-19? Please discuss also the role of the Mediterranean diet in vitamin D levels (PMID: 32429342; PMID: 33509003). This is an important issue relating anti-inflammatory diet and vitamin D levels.
Answer: Thank you. As you suggested, we further expanded the vitamin D section.
Regarding zinc, please refer also to studies using zinc together with hydroxychloroquine in COVID-19.
Answer: Thank you. As you suggested, we also evaluated the synergistic action between zinc and hydroxychloroquine.
Regarding vitamin A, please discuss the potential of hepatotoxicity with its administration.
Answer: Thank you. As you suggested, we discussed about potential of hepatotoxicity with vitamin A administration.
It is important to include immune-boosting nutraceuticals such as curcumin, bromelain (PMID: 33291560, PMID: 33205039, PMID: 33133525)
Answer: Thank you. As you required, we added your suggestions in discussion section.
Reviewer 2 Report
Parisi and colleagues want to summarize, by a review of the literature, the role of nutraceuticals in viral infections and COVID-19 infection, with a special attention to children.
The aim of the work is relevant and interesting, but the manuscript need to improve in ordered to be accepted.
In attached the authors could find a PDF file with my comments highlighted in the text. Moreover below some further comments are reported:
- In general I found that paragraphs don't follow all the same structure. For instance, explanation of the molecule in the immune system; studies that reported this evidence; role and studies in children should be expected in each of the section dedicated to the single molecule. Some of them (ex. Zinc) don't have comments about children. I believe that even when no data on studies on children are found,it should be reported since it is claimed as the aim of the study. The same goes for COVID-19. Moreover for some nutraceutics a lot of information is reported, in some case even too much (not relevant for the aim of the work) while for others, a lot of information is missing. What I expect to find in a review, is a clear presentation on the studies that have been investigated the topic, and I could find it for all the molecules.
- I believe that the paragraph dedicated to vitamine D should be arranged in a clearer way, and I suggest to add some systematic-review/meta-analysis among the listed studies.
- Table and Figures on the contrary are clear and well-done
- I found the Conclusion paragraph not clear, this probably due to a wrong order in which the paragraphs are presented.

Author Response
Reviewer 2
Parisi and colleagues want to summarize, by a review of the literature, the role of nutraceuticals in viral infections and COVID-19 infection, with a special attention to children.
The aim of the work is relevant and interesting, but the manuscript need to improve in ordered to be accepted.
In attached the authors could find a PDF file with my comments highlighted in the text. Moreover below some further comments are reported:
In general I found that paragraphs don't follow all the same structure. For instance, explanation of the molecule in the immune system; studies that reported this evidence; role and studies in children should be expected in each of the section dedicated to the single molecule. Some of them (ex. Zinc) don't have comments about children. I believe that even when no data on studies on children are found,it should be reported since it is claimed as the aim of the study. The same goes for COVID-19. Moreover for some nutraceutics a lot of information is reported, in some case even too much (not relevant for the aim of the work) while for others, a lot of information is missing. What I expect to find in a review, is a clear presentation on the studies that have been investigated the topic, and I could find it for all the molecules.
Answer: Dear reviewer, thank you for your suggestions. We tried to review the manuscript following the same structure for each paragraph: 1) molecule characteristics; 2) actions on the immune system; 3) specific evidences on SARS-CoV2 infections; 4) specific evidences in children. When evidences are lacking, we stated that in the text. We have also edited the manuscript following the corrections suggested by your pdf file.
I believe that the paragraph dedicated to vitamine D should be arranged in a clearer way, and I suggest to add some systematic-review/meta-analysis among the listed studies.
Answer: Thank you. We revised vitamin D section and we expanded it.
Table and Figures on the contrary are clear and well-done
Answer: Thank you. We modified and implemented table 1 with clinical evidence.
I found the Conclusion paragraph not clear, this probably due to a wrong order in which the paragraphs are presented.
Answer: Thank you. The conclusions section focuses heavily on the clinical repercussions of the reported scientific evidence. We tried to revised that paragraph trying to make it clearer.
Round 2
Reviewer 2 Report
I thank the authors to address in a proper way all my comments. They did a lot of effort to improve their manuscript.
I don't have any further comments.